# NEURAL MAXIMUM COMMON SUBGRAPH DETECTION WITH GUIDED SUBGRAPH EXTRACTION

## ABSTRACT

Maximum Common Subgraph (MCS) is defined as the largest subgraph that is commonly present in both graphs of a graph pair. Exact MCS detection is NP-hard, and its state-of-the-art exact solver based on heuristic search is slow in practice without any time complexity guarantee. Given the huge importance of this task yet the lack of fast solver, we propose an efficient MCS detection algorithm, NEURALMCS, consisting of a novel neural network model that learns the node-node correspondence from the ground-truth MCS result, and a subgraph extraction procedure that uses the neural network output as guidance for final MCS prediction. The whole model guarantees polynomial time complexity with respect to the number of the nodes of the larger of the two input graphs. Experiments on four real graph datasets show that the proposed model is $31.78\times$ faster than the exact solver while achieving near-perfect accuracy in MCS detection.

## 1 INTRODUCTION

Graph data are ubiquitous. Due to its flexible and expressive nature, graphs have been used to store data of various domains. In computational biology, atomic networks can represent molecular compounds. In software analysis, program dependence graphs can describe the data and control dependencies. In social science, social networks can represent community structures. Because of graphs' unique ability to capture these data, algorithms tackling novel tasks on graphs across different domains have been gaining increased interest in the representation learning community.

Graph matching, in particular, is a recently popular task with new approaches such as Zanfir & Sminchisescu (2018), Bai et al. (2019a), and Li et al. (2019). These methods either produce a score indicating how much and/or how well two graphs match or a graph alignment indicating how and where two graphs match. The latter case is a far more difficult task than the former one. Current methods perform a special case of graph alignment, namely image matching, where the two input graphs are of spatial structures (ex. pixel grids, object orientation, etc.) To account for more general graphs, we extract the Maximum Common Subgraph (MCS) (Bunke & Shearer, 1998), a widely used metric for graph alignment, and perform matching on the extracted subgraphs.

MCS is a very useful metric to match two graphs in real-world applications. For example, in drug discovery, identifying compounds sharing similar substructures which tend to share similarity properties can dramatically reduce the amount of molecules that need to be manually tested (Ehrlich & Rarey, 2011). In addition to molecular science, MCS also has application values in malware detection (Park et al., 2013), pattern recognition (Solnon et al., 2015), computer-aided circuit design (Djoko et al., 1997; Li et al., 2012), etc. Unfortunately, MCS is NP-hard and, to the best of our knowledge, no existing algorithms tackle this problem from a purely machine learning approach.

We are among the first to tackle graph matching defined by MCS. This is more challenging than image matching or graph similarity computation, because MCS requires the extraction of the largest connected subgraph that is commonly present in both input graphs. This implies that the two extracted subgraphs must not only be contained in both graphs but also be isomorphic to each other. To capture the MCS definition, our proposed model, NEURALMCS, fundamentally changes the way that the representations are learned. Instead of computing similarity all in one step, we introduce an iterative procedure to match nodes one at a time. By selecting nodes successively, we ensure the extracted subgraphs are connected. We utilize subgraph embeddings to perform a stopping condition check, dealing with subgraph isomorphism and ending the procedure when the connected subgraphs

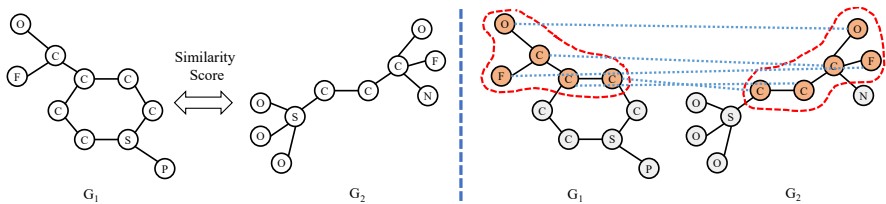

Figure 1: For a graph pair $(\mathcal{G}_1, \mathcal{G}_2)$, previous works (Bai et al., 2019a; 2018; Li et al., 2019) focus on predicting their graph-graph similarity score. In this work, we aim to find the Maximum Common Subgraph (MCS) (circled in red), which requires fine-grained node-node correspondence prediction. It is more useful both due to the application value as described in Section 1 and because of the ***interpretable*** similarity result indicated by the node-node mapping. Node text indicates node labels.

are the largest. By performing this iterative procedure, we both better ensure the isomorphism of the extracted subgraph as well as capture the recurrent relationship between matching pairs of nodes.

We experimentally verify NEURALMCS on four real graph datasets. and show that NEURALMCS can achieve $48.1\times$ runtime gain over state-of-the-art exact MCS computation algorithm MC-SPLIT (McCreesh et al., 2017), and is much more accurate than all the baseline approximate approaches to graph matching.

## 2 PROBLEM DEFINITION AND PRELIMINARIES

### 2.1 PROBLEM DEFINITION

We denote a graph as $\mathcal{G} = (V, E)$ with node set $V$ and edge set $E$. We define as an induced subgraph as $\mathcal{G}' = (V', E')$ where $V' \subseteq V$ and $E' \subseteq E$ and $E'$ preserves all edges between nodes in $V'$ in the original graph $\mathcal{G}$. In other words, for two nodes in $V'$, if there is an edge between them in $\mathcal{G}$, the induced subgraph must also contain the edge. In the rest of the paper, we use the term "subgraph" to refer to "induced subgraph".

In this work, our goal is to detect the Maximum Common Subgraph (MCS) for an input graph pair $(\mathcal{G}_1, \mathcal{G}_2)$. In general, MCS refers to the largest (induced) subgraph that is common to both graphs. In this paper, we make the following qualifications:

- Labeled graph. The nodes of $\mathcal{G}_1$ and $\mathcal{G}_2$ are labeled, and nodes of different labels cannot be matched.
- Connected subgraph. We require the detected subgraph to be connected, which is a common qualification consistent with McCreesh et al. (2017).

Our goal is to detect the MCS for an input graph pair $\mathcal{G}_1, \mathcal{G}_2$. We adopt the state-of-the-art exact solver, MCSPLIT (McCreesh et al., 2017), to provide ground-truth MCS results for a set of training graph pairs. Specifically, MCSPLIT provides not only the nodes that are included in the MCS in both graphs, but also the node-node correspondence in the MCS, as illustrated in Figure 1.

### 2.2 NODE REPRESENTATION LEARNING

Among various node embedding methods, GRAPH MATCHING NETWORKS (GMN) (Li et al., 2019) are recent models designed for graph similarity computation. GMN is based on GRAPH CONVO-LUTIONAL NETWORKS (GCN) (Kipf & Welling, 2016), but in addition to message passing within a graph (intra-graph), GMN explicitly handles inter-graph information passing between the two input graphs.

First, GMN performs explicit cross-graph communication on node embeddings denoted as $\boldsymbol{h}^{(l)}$ where initial node features have $l = 0$. More specifically, cross-graph communication is achieved through the following attention based mechanism: $a_{j \to i} = \frac{\exp(\cos(\boldsymbol{h}_i^{(l)}, \boldsymbol{h}_j^{(l)}))}{\sum_{j'} \exp(\cos(\boldsymbol{h}_i^{(l)}, \boldsymbol{h}_{j'}^{(l)}))}$. Next, a softmax function is applied to the cosine similarity between two node embeddings which makes sure similar node pairs across graphs receive greater attention. This weight is then multiplied with the difference

between cross-graph node embeddings to allow all node pairs in the two graphs to communicate with each other. Finally each node embedding resulted from inter-graph message passing is concatenated with embedding from intra-graph neighborhood aggregation and placed through a MLP or a GRU core to complete a single layer update of a $h_i$.

# 3 THE PROPOSED APPROACH: NEURALMCS

Our proposed approach, NEURALMCS, relies on the learning capacity of the embedding model to generate a good matching matrix for each input graph pair, encoding the likelihood of each node-node pair being included in the MCS and matched to each other. Therefore, the training process aims to learn a matching matrix for each graph pair that is as close to the ground-truth node-node correspondence (as illustrated in Figure 1) as possible.

However, we suppose that a good matching matrix by itself is not enough for an accurate prediction of the MCS, mainly due to the fact that MCS by definition requires the two extracted subgraphs must be isomorphic to each other and both subgraphs must also be connected (see Section 2.1). To satisfy these two requirements, we propose a novel GUIDED SUBGRAPH EXTRACTION (GSE) process that iteratively performs a guided search procedure to enlarge both extracted subgraphs using the matching matrix as guidance. The rest of the section details our proposed approach.

## 3.1 MATCHING MATRIX GENERATION

Our task fundamentally requires the matching between two graphs. Therefore, the ideal node embeddings should receive information from the nodes of both graphs. Thus, we adopt the state-of-the-art node embedding model, GRAPH MATCHING NETWORKS (GMN), as described in Section 2.2. Specifically, we stack $L$ GMN layers on the input node representations to allow inter-graph message passing at multiple scales for sufficient interaction of the two graphs. We denote the final node representations as $\boldsymbol{U}_1 \in \mathbb{R}^{|V_1| \times D^{(L)}}$ and $\boldsymbol{U}_2 \in \mathbb{R}^{|V_2| \times D^{(L)}}$.

To match nodes from the input graphs, we compute the likelihood of matching each node in $\mathcal{G}_1$ to each node in $\mathcal{G}_2$. This likelihood indicates which node pair is most likely to be in the MCS, and should account for cases where some nodes in $\mathcal{G}_1$ do not match any nodes in $\mathcal{G}_2$ (and vice-versa). We naturally encode these likelihoods into a matching matrix, $\boldsymbol{Y} \in [0, 1]^{|\mathcal{G}_1| \times |\mathcal{G}_2|}$.

To compute $\boldsymbol{Y}$, one can simply calculate the dot product between the node embeddings, $\boldsymbol{U}_1 \boldsymbol{U}_2^\top$. However, this resulting matrix is simply the similarity score between each pair of nodes in the two graphs, which is in the range of $(-\inf, +\inf)$ and requires further processing to reflect the probability of the node pair matched and being included in the MCS.

As this matrix encode the general similarity between nodes, we denote it as the similarity matrix, $\boldsymbol{X}$, and further normalize it to obtain $\boldsymbol{Y}$. Specifically, we perform the following transformations.

To find the likelihood of each node in one graph matching each node in the other graph, we apply column-wise and row-wise normalization on $\boldsymbol{X}$.

$$\tilde{p}_{coln}(i,j) = \frac{e^{X_{ij}}}{\sum_k e^{X_{ik}}}, \quad \tilde{p}_{row}(i,j) = \frac{e^{X_{ij}}}{\sum_k e^{X_{kj}}} \quad (1)$$

To allow for some nodes to go unmatched, we multiply these likelihoods by a value encoding the overall matching score of a node in one graph to nodes in the other graph.

$$p_{coln}(i,j) = \sigma\left(\frac{\sum_j X_{ij}}{|V_2|}\right) \cdot \tilde{p}_{coln}(i,j), \quad p_{row}(i,j) = \sigma\left(\frac{\sum_i X_{ij}}{|V_1|}\right) \cdot \tilde{p}_{row}(i,j) \quad (2)$$

We consider both row-wise and column-wise normalization by taking the average of both scores, $\tilde{p}(i,j) = \frac{p_{coln}(i,j) + p_{row}(i,j)}{2}$, and, to ensure we only select nodes with the same labels, we mask out node pairs with different labels to form $\boldsymbol{Y}$.

## 3.2 LEARNING OF NEURALMCS

Although there exist multiple points where we can apply our loss function (before, during, or after GSE), we found that applying binary cross entropy loss as early as possible allows for GMN to

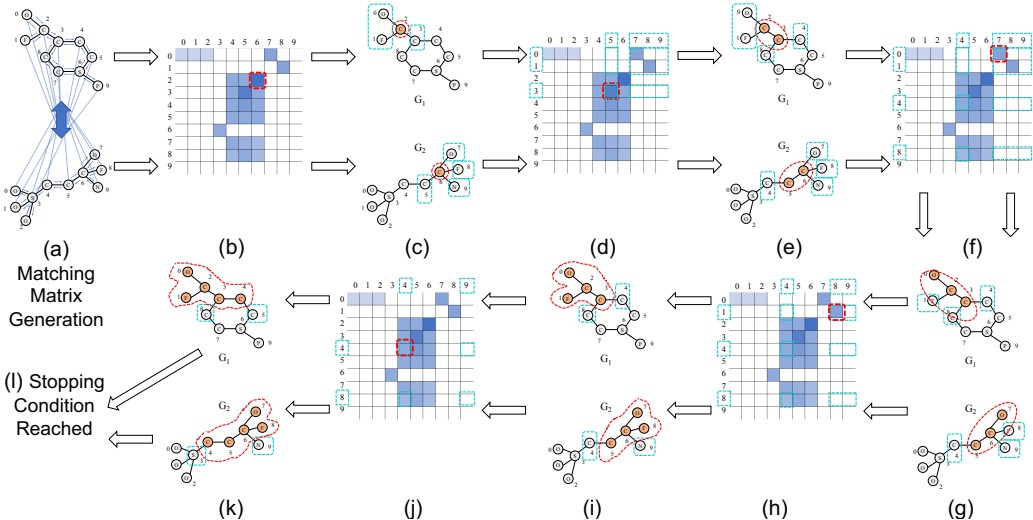

(a) Matching Matrix Generation

(l) Stopping Condition Reached

(b) (c) (d) (e) (f)

(k) (j) (i) (h) (g)

Figure 2: A detailed illustration of NEURALMCS using the example graph pair in Figure 1. For the input graph pair $(\mathcal{G}_1, \mathcal{G}_2)$, the NEURALMCS first generates a matching matrix encoding the likelihood of each pair of node being matched ((a) and (b)). The GUIDED SUBGRAPH EXTRACTION (GSE) process uses the matching matrix as follows: NEURALMCS selects the initial pair to be included in the predicted MCS, node 2 in $\mathcal{G}_1$ and node 6 in $\mathcal{G}_2$ ((b) and (c)). Next, NEURALMCS sets its search frontier as the neighbors of the selected nodes, circled in dashed blue lines. By selecting the pair with the largest matching score which still preserves subgraph isomorphism, node 3 in $\mathcal{G}_1$ and node 5 in $\mathcal{G}_2$, the extracted subgraphs in both graphs grow to size 2 ((d) and (e)). The procedure continues until a stopping condition is reached, which is detailed in Section 3.3.

receive a better learning signal. For this reason, our loss function acts directly on the matching matrix.

Specifically, we design the following loss function to train the model $L(\boldsymbol{Y}) = -\sum_i \sum_j \frac{I_{ij} log(Y_{ij})}{|V_1| \cdot |V_2|}$ where $Y_{ij}$ denotes our predicted matching matrix and $I_{ij}$ denote an indicator value of whether the node $i$ matches node $j$ in the ground truth.

Since for a given graph pair, there can be multiple correct MCSs of equal size, we also employ multiple choice learning (Guzman-Rivera et al., 2012; Li et al., 2018). This is done by ensembling various copies of the model and propagating the loss function only through the copy which achieves the lowest loss, as shown by the following: $L = min_{\boldsymbol{Y} \in \mathcal{Y}}(L(\boldsymbol{Y}))$.

## 3.3 GUIDED SUBGRAPH EXTRACTION (GSE)

Given a matching matrix $\boldsymbol{Y}$ encoding the matching likelihood for all node pairs between $\mathcal{G}_1$ and $\mathcal{G}_2$, we propose the following GUIDED SUBGRAPH EXTRACTION (GSE) process. It starts by finding the most likely pair, and iteratively expands the extracted subgraphs by selecting one more node pair at a time. The procedure stops once the addition of any additional pair would lead to non-isomorphic subgraphs. In summary, the proposed algorithm is shown in Algorithm 1. A detailed illustration using an example pair from Figure 1 is shown in Figure 2.

The GSE algorithm internally maintains a binary assignment matrix $\boldsymbol{T}$ which will be the final output indicating the predicted MCS, and a masking matrix $\boldsymbol{M}$, which is used to mask and select entries of the matching matrix $\boldsymbol{Y}$. Both matrices are of the same dimension as $\boldsymbol{Y}$. $\boldsymbol{T}$ is initialized to all zeros, indicating no selected nodes, i.e. extracted subgraphs are zero-size. $\boldsymbol{M}$ is initialized to all ones, since NEURALMCS may select any node pair for its initial subgraph.

To select the most likely node matching, the algorithm decides which pair of nodes should be selected (see Section 3.3.1 for details) by order of their matching scores. This involves checking whether the selection of the pair would result in two isomorphic subgraphs or not. Only if so, the algorithm would select the pair (circled in red in Figure 1) and include it in the predicted MCS by updating $T_{i^\dagger, j^\dagger}$ to be 1. Once a new node pair is included, GSE updates the mask $\boldsymbol{M}$ to reflect the

---

**Algorithm 1** GUIDED SUBGRAPH EXTRACTION (GSE)

1: **Input:** $\{A_1, A_2\}$, $Y$, node embeddings $\{U_1, U_2\}$, $\epsilon$.
2: **Output:** Assignment matrix $T$.
3: Initialize $T \leftarrow 0 * Y$. \\ initialize to all zeros
4: Initialize $M \leftarrow 1 * Y$. \\ initialize to all ones
5: Initialize update $\leftarrow$ True
6: **while** update = True
7:    $I \leftarrow$ sorted indices by highest to lowest value of elements in $\mathbf{M} \odot \mathbf{Y}$
8:    **for** node pair $(i^\dagger, j^\dagger) \in \mathbf{I}$
9:       update $\leftarrow$ False
10:      Compute subgraph embeddings $\boldsymbol{w}_1, \boldsymbol{w}_2$ via Equation 4.
11:      **if** $\|\boldsymbol{w}_1 - \boldsymbol{w}_2\|_2 \le \epsilon$ \\ subgraph isormorphism check
12:         Select the found pair $(i^\dagger, j^\dagger)$ by updating $T_{i^\dagger, j^\dagger} \leftarrow 1$.
13:         Expand the search frontier by updating the mask $M$ via Equation 6.
14:         update $\leftarrow$ True
15:         **break**

---

search frontier for the next iteration (see Section 3.3.2 for details) and proceeds to the next iteration. Since the MCS by definiton requires the extracted subgraphs to be connected graphs, we define the search frontier as the first-order neighboring nodes of the current selected nodes. This guarantees the final predicted MCS satisfies the connectivity constraint described in Section 1.

### 3.3.1 SUBGRAPH ISOMORPHISM CHECK

To decide whether node pair $(i^\dagger, j^\dagger)$ from the search frontier can be selected or not, we check if the inclusion of node $i^\dagger$ in $\mathcal{G}_1$ and $j^\dagger$ in $\mathcal{G}_2$ would result in two isomorphic subgraphs. To do this, we compute the two subgraph-level embeddings, $\boldsymbol{w}_1$ and $\boldsymbol{w}_2$ and check if their Euclidean distance is greater than a hyperparameter threshold $\epsilon$. A more detailed discussion can be found in Appendix H and I.

Computing the subgraph-level embeddings, however, involves more than a simple aggregation of node embeddings from $U_1$ or $U_2$. This is because for subgraph isomorphism check, we are only concerned with nodes included in the subgraph. The nodes outside the selected subgraph should not affect the nodes inside. However, during node embedding generation, both intra- and inter-graph message passing are performed, resulting in $U_1$ and $U_2$ containing unwanted information.

Inspired by the Weisfeiler-Lehman (WL) algorithm for approximate graph isomorphism test (Shervashidze et al., 2011), as well as the connection between the WL algorithm and the GCNs, we recompute the node embeddings by only aggregating neighboring nodes that are included in the current predicted MCS indicated by $T$:

$$
\begin{aligned}
\boldsymbol{U}_1^{(k+1)} &= \boldsymbol{A}_1(\boldsymbol{v_1} \odot \boldsymbol{U}_1^{(k)}), \\
\boldsymbol{U}_2^{(k+1)} &= \boldsymbol{A}_2(\boldsymbol{v_2} \odot \boldsymbol{U}_2^{(k)}),
\end{aligned}
\tag{3}
$$

where $A$ denotes the adjacency matrix, $\boldsymbol{v}_1 \in \{0, 1\}^{|V_1|}$ and $\boldsymbol{v}_2 \in \{0, 1\}^{|V_2|}$ are defined as the row and column summation[1] of $T$, i.e. $v_{1i} = \sum_{j=1}^{|V_2|} T_{i,j}, v_{2j} = \sum_{i=1}^{|V_1|} T_{i,j}$, followed by setting $v_{1i^\dagger}$ and $v_{2j^\dagger}$ to 1, and $\boldsymbol{v} \odot \boldsymbol{U}$ denotes element-wise multiplication, i.e. $(v \odot U)_{i,j} = v_i U_{i,j}$. For each graph, this performs message passing between nodes in the extracted subgraph plus the node $i^\dagger$ (for $\mathcal{G}_1$) or $j^\dagger$ (for $\mathcal{G}_2$). All the edges between these nodes are involved via the adjacency matrices $A_1$ and $A_2$, ensuring the subgraph is an induced subgraph required by the MCS definition as mentioned in Section 2.1.

The above updates are performed for $K$ times to yield the final subgraph node embeddings denoted as $\boldsymbol{U}_1^{(K)} \in \mathbb{R}^{|V_1| \times D^{(K)}}$ and $\boldsymbol{U}_2^{(K)} \in \mathbb{R}^{|V_2| \times D^{(K)}}$. We finally compute the two subgraph-level

---

[1]By design of GSE, $T$ is guaranteed to be an assignment matrix, i.e. there is at most one 1 for each row and column (see Section 3.3.2). Thus $v_1$ and $v_2$ are binary vectors indicating whether a node is included in the subgraphs.

embeddings whose Euclidean distance is computed for apporximate subgraph isomorphism test:

$$\begin{aligned} \boldsymbol{w}_1 &= \sum_{i=1}^{|V_1|}(\boldsymbol{v_1} \odot \boldsymbol{U}_{1i,:}^{(K)}), \\ \boldsymbol{w}_2 &= \sum_{j=1}^{|V_2|}(\boldsymbol{v_2} \odot \boldsymbol{U}_{2j,:}^{(K)}), \end{aligned} \tag{4}$$

### 3.3.2 SEARCH FRONTIER EXPANSION

As the search process finds more node pairs, the mask $\boldsymbol{M}$ reflects the search frontier, i.e. the candidate node pairs for the next iteration to select from. Specifically, $M_{i,j}$ denotes whether the node pair $(i, j)$ should be a candidate pair, and the sorting for node pairs is performed on $\boldsymbol{M} \odot \boldsymbol{Y}$.

Once node pair $(i^{\dagger}, j^{\dagger})$ is selected by passing the check described in Section 3.3.1, GSE updates the search frontier by first obtaining two binary vectors indicating the neighbors of the selected nodes. Since each row of the adjacency matrix indicates the neighbors of the nodes of a particular graph, and we need to include neighbors of all the selected nodes, we first perform the following aggregation of rows of the adjacency matrix:

$$\begin{aligned} \boldsymbol{p}_1 &= (\sum_{i=1}^{|V_1|} \boldsymbol{v}_1 \odot \boldsymbol{A}_{1i,:}) \odot (1 - \boldsymbol{v}_1), \\ \boldsymbol{p}_2 &= (\sum_{j=1}^{|V_2|} \boldsymbol{v}_2 \odot \boldsymbol{A}_{2j,:}) \odot (1 - \boldsymbol{v}_2). \end{aligned} \tag{5}$$

The $(1 - \boldsymbol{v})$ term is for excluding the nodes that are already selected, which is key for ensuring that the next step does not select nodes that are already selected. This further guarantees that the final $\boldsymbol{T}$ matrix is an assignment matrix as required by Equation 3.

To obtain a binary indicator vector for the neighbors of selected nodes, we apply an element-wise indicator function, checking whether each entry of $\boldsymbol{p}_1$ and $\boldsymbol{p}_2$ is greater than zero, yielding $\boldsymbol{q}_1 \in \{0,1\}^{|V_1|}$ and $\boldsymbol{q}_2 \in \{0,1\}^{|V_2|}$. Then we can obtain the updated mask $\boldsymbol{M}$ via

$$\boldsymbol{M} = \boldsymbol{q}_1 \otimes \boldsymbol{q}_2, \tag{6}$$

where $\otimes$ denotes the dyadic product of two vectors. This allows the selection to choose from the pairs formed by the nodes indicated by $\boldsymbol{q}_1$ and $\boldsymbol{q}_2$.

### 3.4 OVERALL TIME COMPLEXITY

Each GMN layer involves the message passing for all node-node pairs, whose time complexity is quadratic with respect to the number of nodes. The matching matrix computation computes the dot product for all node-node pairs. The GSE process selects one node pair each time, and expands the search frontier by reaching to neighbors of selected nodes. Thus, GSE in the worst case reaches out to all nodes of the smaller of the two input graphs, and each GSE step involves a quadratic mask computation. Therefore, NEURALMCS runs in $O(s * |V_1| * |V_2| * log(|V_1| * |V_2|))$ time, where $s$ is the size of the predicted MCS, which in the worst case is $\min(|V_1|, |V_2|)$.

## 4 EXPERIMENTS

We evaluate NEURALMCS on its accuracy and efficiency against the state-of-the-art exact MCS solver, MCSPLIT (McCreesh et al., 2017), and current machine learning approaches (Zanfir & Sminchisescu, 2018; Wang et al., 2019a; Velickovic et al., 2018; Li et al., 2019), on four real-world datasets, AIDS, LINUX, IMDB, and REDDIT from a diverse range of domains. Our baselines incldude the state-of-art MCS computation algorithm, MCSPLIT (McCreesh et al., 2017), IMAGE-GMN (Zanfir & Sminchisescu, 2018), IMAGE-PCA (Wang et al., 2019a), BASIC-GAT (Velickovic et al., 2018), and BASIC-GMN (Li et al., 2019). Appendix A and B give more details.

### 4.1 EVALUATION METRICS

For accuracy, we evaluate an exact and a soft metric:

$$\text{Exact } \% = \frac{\sum_i^N C(S_{i1}, S_{i2}) \cdot \mathbf{1}_{(|S_{i1}|=|\text{MCS}_i|)}}{N} \tag{7}$$

$$\text{Soft } \% = \frac{\sum_i^N C(S_{i1}, S_{i2}) \cdot \frac{|S_{i1}| + |S_{i2}|}{2|\text{MCS}_i|}}{N} \tag{8}$$

where $N$ is the number of graph pairs in the testing set; $S_{i1}$ and $S_{i2}$ are the subgraphs extracted from the first and second graph, respectively; $C(\cdot, \cdot)$ is a function that returns 1 if the two input graphs are isomorphic to each other and 0 otherwise; $\mathbf{1}$ is the indicator function that returns 1 when the condition is true and 0 otherwise; $|\text{MCS}_i|$ is the true MCS size. For efficiency, we evaluate the average running time across graph pairs.

## 4.2 RESULTS

We see that in terms of accuracy, NEURALMCS improves current methods of subgraph extraction substantially, especially on the AIDS and LINUX datasets, where NEURALMCS performs close to the ground truth solver. For IMDB and REDDIT, the performance is also drastically higher than the baselines. Most interestingly, we find that, when the model does not find the exact ground-truth solution, it still extracts high-quality subgraphs with sizes on average above 90% of the true MCS size, where all graphs are isomorphic (see Appendix E). The poor performance of computer vision baselines IMAGE-GMN and IMAGE-PCA suggests that the assumptions for image matching do not work for MCS based graph matching very well. Appendix F provides a thorough analysis of the performance boost from each component of the proposed model compared with various alternative designs. Compared against the 2 basic models, we see that both the representation learning scheme and the extraction strategy greatly enhance the performance of the model. The exact accuracy results can be seen in Table 1.

Table 1: Exact % and Soft % accuracy metrics across four real graph datasets. All methods have been adapted for the MCS detection task. MCSPLIT is the ground-truth MCS solver labeled with *.

| Method | AIDS | | LINUX | | IMDB | | REDDIT | |
|---|---|---|---|---|---|---|---|---|
| | Exact % | Soft % | Exact % | Soft % | Exact % | Soft % | Exact % | Soft % |
| MCSPLIT * | 100.000 | 100.000 | 100.000 | 100.000 | 100.000 | 100.000 | 100.000 | 100.000 |
| IMAGE-GMN | 0.033 | 0.033 | 19.790 | 19.790 | N/A | N/A | 20.261 | 20.261 |
| IMAGE-PCA | 0.229 | 0.229 | 22.693 | 22.693 | 38.582 | 38.582 | 33.987 | 33.987 |
| BASIC-GAT | 11.013 | 22.199 | 41.135 | 53.070 | 47.462 | 51.680 | 37.908 | 50.196 |
| BASIC-GMN | 12.488 | 19.256 | 48.082 | 52.590 | 55.273 | 62.738 | 43.137 | 51.782 |
| NEURALMCS | **98.525** | **99.626** | **99.674** | **99.955** | **97.235** | **99.613** | **96.078** | **99.562** |

In terms of efficiency, NEURALMCS is $31.78\times$ faster than the ground-truth solver averaged across the four datasets, while being slightly slower than the baselines.

However, as shown in Table 1, the baseline methods fail to consider the connectivity and subgraph isomorphism constraints during maximum common subgraph extraction resulting in worse accuracy. The exact performance results can be seen in Figure 3.

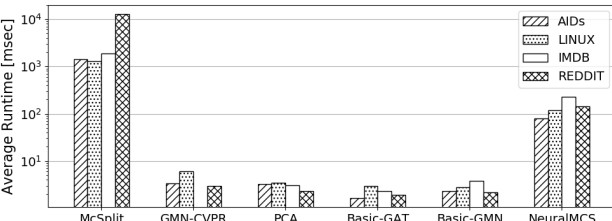

Figure 3: Running time comparison.

## 4.3 CASE STUDY

The iterative expansion ensures that extracted subgraphs satisfy the connectivity constraint. The representation learning component ensures that we select the correct node at each step. As seen in Figure 4, for the AIDS dataset, the learnable matching matrix helps NEURALMCS select good nodes, and, for the IMDB dataset, the iterative procedure indeed helps our model stop once adding additional nodes would break the isomorphism constraint. More importantly, our model is also able to solve both the subgraph isomorphism and graph matching problems. Additional case studies and analysis can be found in Appendix J.

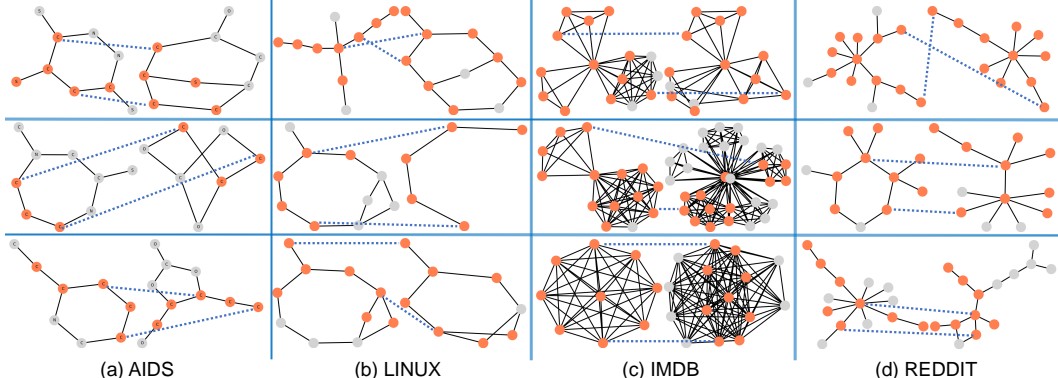

(a) AIDS  (b) LINUX  (c) IMDB  (d) REDDIT

Figure 4: Case study: Three MCS examples from each dataset. AIDS has node labels as node text. For clarity, we draw only two node-node correspondences for each example, represented as dashed lines between nodes in the two graphs.

## 5 RELATED WORK

Graph isomorphism is a classic problem that goes back to at least the early work by Sussenguth (1965) and Corneil & Gotlieb (1970). Recently Babai (2016) shows that graph isomorphism can be solved in quasipolynomial time. However, graph isomorphism is only concerned with whether two graphs are exactly the same or not, which may be too rigid in real-world applications. For example, what if two graphs are not isomorphic? In such scenario, a more useful output can be the similarity (or distance) between them, defined by metrics such as Graph Edit Distance (GED) (Bunke, 1983), Maximum Common Subgraph (MCS) (Bunke & Shearer, 1998; Bunke, 1997).

In this work, we focus on the more challenging task of graph matching, specifically the matching of two graphs defined by the MCS metric. The MCS detection problem is NP-hard, and remain computationally challenging. Existing works use constraint programming (Vismara & Valery, 2008; McCreesh et al., 2016), branch-and-bound (McCreesh et al., 2017; Liu et al., 2019), mathematical programming (Bahiense et al., 2012), reduction to maximum clique detection (Levi, 1973; McCreesh et al., 2016), etc. MCS has many definitions tailored to different graph types and application specifics (McCreesh et al., 2017), and is a domain-agnostic metric to compare graphs in a detailed way, so it has occurred widely in applications such as graph database systems (Yan et al., 2005), cloud computing platforms (Cao et al., 2011), etc.

The term "graph matching" is very general and has been adopted by many recent works: (1) Graph similarity computation (Bai et al., 2019a; Li et al., 2019) aims to output a similarity score for an input graph pair, which can be defined by the MCS metric but less challenging due to the score output. (2) Image matching is a classic task in computer vision with traditional approaches based on quadratic assignment problem solving (Zhou & De la Torre, 2012; Yu et al., 2018) and more recent methods using neural networks (IMAGE-GMN (Zanfir & Sminchisescu, 2018) and IMAGE-PCA (Wang et al., 2019a)). (3) The two-graph alignment (Heimann et al., 2018; Xu et al., 2019) problem deals with the alignment of two general structured graph objects. However, the alignment is typically not defined by domain-agnostic metrics such as GED or MCS. There is currently no such methods learning from ground-truth graph pairs to the best of our knowledge.

## 6 CONCLUSION

In this paper, we propose a neural network approach to solve the NP-hard problem, Maximum Common Subgraph (MCS) detection, in an approximate and accurate way. For an input graph pair, our model NEURALMCS computes the likelihood of each pair of nodes being included in the MCS and matched, which is used by the proposed GUIDED SUBGRAPH EXTRACTION (GSE) algorithm to iteratively include more and more nodes in the predicted MCS. The whole model runs in polynomial time complexity, and experimental results on four real graph datasets demonstrate that NEURALMCS is 31.78× faster than the exact solver whole achieving very good accuracy compared to a series of strong approximate graph matching baseline approaches.

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

## A    DATASET DESCRIPTION

We run experiments on 4 diverse different real world datasets coming from the chemical, programming language, and social network domains. For each dataset, we split the pairs into training, validation, and testing sets in the ratio of 6:2:2 such that none of the training, validation, or testing sets share any common graphs. For each dataset, we either one-hot encode node labels, if the dataset has node labels, or provide the same initial encoding, if the dataset does not have node labels. The code and the datasets have been published to this anonymous link: `https://github.com/openpublicforpapers/NeuralMCS`.

### A.1    AIDS

AIDS is a dataset of antivirus screen chemical compounds, coming from the Developmental Therapeutics Program at NCI/NIH[2]. The AIDS dataset has been used by various works in graph matching (Zeng et al., 2009; Wang et al., 2012; Zheng et al., 2013; Zhao et al., 2013; Liang & Zhao, 2017; Bai et al., 2019a). These graphs consist of labeled nodes and unlabeled edges, where nodes represent chemical elements (ex. Carbon, Nitrogen, Chlorine, and etc.) and edges represent bonds between atoms. There are a total of 700 graphs, from which we sample 29610 graph pairs. The average graph size is 8.664 with the largest graph having 10 nodes.

### A.2    LINUX

LINUX is a dataset of program dependence graphs (PDG) describing individual functions generated from the Linux kernel (Wang et al., 2012). These graphs consist of unlabeled nodes and unlabeled edges, where nodes represent statements and edges represent control flow between statements. There are a total of 1000 graphs, from which we sample 60114 graph pairs. The average graph size is 7.591 and the largest graphs have 10 nodes.

### A.3    IMDB

IMDB is a dataset of ego-networks of movie actors/actresses from the IMDB website (Yanardag & Vishwanathan, 2015). This dataset has been used by various works in graph classification (Zhang & Chen, 2019; Bai et al., 2019b). These graphs consist of unlabeled nodes and unlabeled edges, where nodes represent actors/actresses and edges represent whether they have had collaborations. There are a total of 1500 graphs, from which we sample 135702 graph pairs. The average graph size is 12.981 and the largest graphs have 89 nodes.

---

[2] `https://wiki.nci.nih.gov/display/NCIDTPdata/AIDS+Antiviral+Screen+Data`

## A.4  REDDIT

REDDIT is a dataset of online dicussion networks from the Reddit online discussion website (Yanardag & Vishwanathan, 2015). These graphs consist of unlabeled nodes and unlabeled edges, where nodes represent users and edges represent whether users have responded to eachother's comments. There are a total of 7112 graphs, from which we sample 3556 pairs. The average graph size is 11.8 nodes and the largest graph has 16 nodes.

## B  BASELINES

We evaluate NEURALMCS against the state-of-the-art solver, MCSPLIT, which uses heuristics and the branch and bound algorithm to achieve efficient computation for MCS. While the fastest among exact solvers, this method still has, in the worst case, exponential time complexity (McCreesh et al., 2017). For this reason, in practice, its running time is much slower than the other approximate baselines. However, we still use MCSPLIT to generate ground-truth MCS results and it only takes a few days on a standard CPU server with multi-threading to handle all the ground-truth result generation.

The representation learning baselines are adapted from recent graph matching techniques, specifically IMAGE-GMN (Zanfir & Sminchisescu, 2018) and IMAGE-PCA (Permutation Loss and Cross-graph Affinity) (Wang et al., 2019a). Both these methods utilize similarity scores and normalization to perform graph matching. As IMAGE-GMN uses CNN layers to form their node embeddings from images (with techniques such as Delaunay triangulation (Lee & Schachter, 1980)), we adapt the model to our task by replacing these layers with 3 GAT layers, consistent with NEURALMCS. IMAGE-PCA uses a node embedding mechanism similar to GMN, not requiring further adaptation. As the loss functions for both these methods were designed for image graphs, we alter their loss functions to binary cross entropy loss (the same as NEURALMCS).

We conduct the extraction process by first removing all nodes where its corresponding summation of similarity scores across rows or columns falls below a tuneable threshold (the values of which can be found in Appendix C) in the matching matrix, then selecting an equal number of nodes in both graphs. The latter is done by maximizing the similarity scores of uniquely selected node pairs through applying the Hungarian Algorithm (Kuhn, 1955) on the remaining nodes of the matching matrix. This is because the extracted subgraphs, after just thresholding, may be of different sizes due to thresholding. The extra procedure ensures that we extract subgraphs of equal sizes and in addition, the Hungarian Algorithm yields one-to-one node-node correspondences for the extracted subgraphs. However, these methods do not necessarily result in isomorphic subgraphs.

We also compare against two basic models, BASIC-GAT (Velickovic et al., 2018) and BASIC-GMN (Li et al., 2019), which use 3 GAT and GMN layers respectively to encode the initial node features, followed by similarity score computation. These baselines uses a simpler normalization scheme by appling element-wise sigmoid function ($y = 1/(1 + e^{-x})$) to the matching matrix, and directly feed the similarity matrix to the same loss function as defined by NEURALMCS. BASIC-GAT and BASIC-GMN use the same extraction method as the graph matching baselines.

With exception to MCSPLIT, all these baselines run in polynomial time complexity with respect to the number of nodes in the two input graphs.

## C  PARAMETER SETTINGS

For our model, we utilize 3 layers of GMN with 64 dimensions each for the initial embedding. We use $\text{ReLU}(x) = \max(0, x)$ as our activation function. We fix the number of outputs for multiple choice learning to 10 on the AIDS, LINUX, and REDDIT datasets, and to 3 on the IMDB dataset. To evaluate the efficacy of multiple choice learning, we also use the same technique and settings for BASIC-GAT and BASIC-GMN. We set $\epsilon$ to $10^{-4}$, to account for numerical errors during floating point arithmetic. For the extraction procedure in our baselines, we set the threshold to 0.5 on the AIDS, LINUX, and REDDIT datasets, and to 0.7 on the IMDB dataset. We ran all experiments with Intel i7-6800K CPU and one Nvidia Titan GPU. For training, we set the batch size to 64, the learning rate to 0.001, the number of iterations to 5000, and use the Adam optimizer (Kingma & Ba,

2015). All experiments were implemented with the PyTorch and PyTorch Geometric libraries (Fey & Lenssen, 2019).

For more on dataset setup and training/validation/testing splits, please refer to Appendix A. For more on baseline setups, including their extraction procedure, please refer to Appendix B. For more on the extraction procedure of NEURALMCS, please refer to the main text.

## D   DETAILS ON EVALUATION PROCEDURE

As mentioned in Section 4.1, for a graph pair $i$ in the test set, we first check whether the predicted MCS satisfies the MCS constraints or not. In this section we give more details.

Denote the extracted subgraphs from the $i$-th graph pair as $S_{i1}$ and $S_{i2}$, respectively. The MCS constraint satisfaction check includes the following steps: First, we check if $S_{i1}$ and $S_{i2}$ are both connected (i.e. no isolated components); Second, we check if $S_{i1}$ and $S_{i2}$ are isomorphic to each other. Since exact isomorphism checking may take a long time in practice, we set a timeout for exact isomorphism checking, and when timeout happens, we switch to an approximate checker[3]. The timeout as well as the whole procedure is applied across all the methods to ensure fair comparison.

When we generate $S_{i1}$ and $S_{i2}$ for NEURALMCS in the first place, we would take the induced subgraph, satisfying the inductivity constraint mentioned in Section 2.1. Specifically, we use the output assignment matrix $T$ and check if a node has a 1 in its corresponding row (for $\mathcal{G}_1$) or column (for $\mathcal{G}_2$) in $T$ to decide whether it is selected in the predicted MCS.

Notice that by checking the size of the extracted subgraphs against the ground-truth MCS size (Equation 7 and 8), we allow equal-sized MCS results to be evaluated correctly, since for many graph pairs there are more than one correct MCS result.

Here we give more insights behind the two metrics. The intuition behind the hard metric is that we only check whether the the extracted subgraphs are strictly equal to the size of the true MCS or not. However, this is too strict and does not reveal too much information about the model performance when the extracted subgraphs are not the same size as the true MCS. An easy measure would be to directly report on the predicted MCS size, but this does not take into account non-isomorphic subgraphs. The soft metric accounts for both these issues by checking the fraction of the predicted MCS size over the true MCS size only for isomorphic subgraphs. Notice that if the predicted MCS is even larger than the true MCS, the $C(\cdot, \cdot)$ function will return 0 because it is not possible for the subgrpahs to be both larger than the true MCS (generated by the exact solver MCSPLIT) and isomorphic.

## E   RESULT ANALYSIS

In addition to Exact % and Soft % metrics, we may also evaluate our model on the percentage of extracted subgraphs which are isomorphic (Iso %).

Following the same notation used to define Exact % and Soft %, we define Iso % as follows:

$$\text{Iso } \% = \frac{\sum_i^N C(S_{i1}, S_{i2})}{N} \quad (9)$$

As seen in Table 2, NEURALMCS explicitly encodes the isomorphism constraint into the stopping condition. This is one reason why NEURALMCS is able to achieve high accuracy, especially in Soft % in Table 1. Notice, our method does not guarantee exclusion of false-positives (predicting isomorphic when not isomorphic), as it is possible for 2 differently structured graphs, with different node embeddings, to have the same subgraph embedding once all the node embeddings are aggregated. Because these situations are relatively rare (both graphs would need to have the same subgraph embedding to floating point precision), NEURALMCS is extract isomorphic subgraphs 100% of the

---

[3]https://networkx.github.io/documentation/networkx-2.1/reference/
algorithms/generated/networkx.algorithms.isomorphism.could_be_isomorphic.
html

time on the provided datasets. Our model does guarantee exclusion of false-negatives, as, if two graphs are isomorphic, our embedding propagation methodology must produce the same subgraph embeddings.

Table 2: Iso % accuracy metric across four real graph datasets. All methods have been adapted for the MCS detection task. Dataset descriptions and details can be found in Appendix A.

| Method | AIDS | LINUX | IMDB | REDDIT |
|---|---|---|---|---|
| MCSPLIT * | 100.000 | 100.000 | 100.000 | 100.000 |
| IMAGE-GMN | 0.033 | 19.790 | N/A | 20.261 |
| IMAGE-PCA | 0.229 | 22.693 | 38.582 | 33.987 |
| BASIC-GAT | 80.039 | 82.269 | 99.365 | 73.856 |
| BASIC-GMN | 91.183 | 97.230 | 96.924 | 95.425 |
| NEURALMCS | **100.000** | **100.000** | **100.000** | **100.000** |

# F ABLATION STUDY

We first form a matching matrix then extract a subgraph guided by this matrix. To form the matching matrix, we utilize representation learning to make node embeddings; compute $X$ using similarity scores from node embeddings (Section 3.1); compute $Y$ through normalization of $X$ (Section 3.1). To perform extraction, we utilize the GUIDED SUBGRAPH EXTRACTION method proposed (Section 3.3).

We perform more in-depth ablation studies to show the importance of each component whose results are shown in Table 3.

Table 3: Abaltion study results on AIDS. The numbers are the "Exact %" defined in Section 4.1.

| Matching Matrix Computation | Subgraph Extraction Strategy | | |
|---|---|---|---|
| | GSE | Threshold | Threshold + LSAP |
| GMN + Our Normalization | 98.525 (NEURALMCS) | 25.795 | 17.076 |
| GAT + Our Normalization | 98.525 | 26.057 | 17.339 |
| DGCNN + Our Normalization | 96.657 | 22.091 | 13.045 |
| GMN + Sigmoid | 97.083 | 10.521 | 12.488 (BASIC-GMN) |
| GMN + Sinkhorn Softmax | 60.439 | 12.488 | 0.197 |

## F.1 ON THE IMPORTANCE OF GMN

GAT + Our Normalization and DGCNN + Our Normalization use GAT (Velickovic et al., 2018) and DGCNN (Wang et al., 2019b) used in Wang & Solomon (2019) respectively to perform node embeddings. Interestingly, when fed into our proposed GSE step, their exactly solved percentages are quite close to GMN, which is close to perfectly detecting the MCSs for all the testing pairs. Even with simpler thresholding based subgraph extraction strategies (see Section F.3 below for details), their performances are still similar to (or even better than) GMN. This seems to suggest that the choice of node embedding methods does not appear to influence the performance much when our proposed GSE strategy is used.

It should be noted that earlier we used a simpler GSE strategy which always selected the node pair with the largest matching score in $Y$ in during search frontier expansion. This model did not check all the possible node pairs by sorting the node matching scores and iterate through these pairs for consideration of being selected in the MCS prediction. When tested on this simpler GSE strategy, GMN indeed performed approximately $4.0\%$ better than GAT and DGCNN.

In summary, the choice of node embedding representation methods does not influence the performance too much, and very good accuracy can be obtained when the proposed normalization and GSE methods are both used.

### F.2    ON THE IMPORTANCE OF NORMALIZATION

GMN + Sigmoid (BASIC-GMN) uses sigmoid normalization on each individual element of the $X$ matrix, instead of our normalization scheme (Section 3.1; Equations 1 and 2) to obtain $Y$. As sigmoid treats each node-node pair in $Y$ as independent (an incorrect assumption), we see that its performance worse than our proposed normalization scheme, especially when the threshold or threshold + LSAP strategies are used. However, similar to our findings in Section F.1, when our proposed GSE strategy is used, GMN + Sigmoid performs only slightly worse than NEURALMCS, which further confirming the usefulness of the proposed normalization scheme and the GSE method.

GMN + Sinkhorn Softmax uses successive row- and column-wise softmax normalization (softmax to ensure that the matching matrix $Y$ is in the range (0,1)) on the $X$ matrix (similar to the Sinkhorn algorithm (Knight, 2008) used in IMAGE-PCA for image matching[4]) instead of our normalization scheme. As softmax does not explicitly allow nodes to go unmatched (Section 3.1), as dictated by the MCS definition, we see that our normalization procedure performs much better.

As mention in Section 4.2, the assumptions made for image matching do not naturally transfer to and work well for the MCS detection task. In fact, with the simple element-wise sigmoid normalization on $X$, the performance is much better than the Sinkhorn normalization technique. The iterative row- and column-wise normalization on $X$ is not a good choice for the task of MCS where nodes can remain unmatched with low scores in the final $Y$.

### F.3    ON THE IMPORTANCE OF GSE

For each Matching Matrix Computation method, we run 3 different subgraph extraction strategies: GSE, thresholding, and thresholding + LSAP (Linear Sum Assignment Problem which we use the Hungarian algorithm (Kuhn, 1955) to solve, described in Appendix B).

For thresholding, for each of the two graphs, we select the nodes whose probabilities of being included in the MCS are greater than a tunable threshold, yielding two subgraphs. We calculate such probabilities by taking the summation of rows and columns of the matching matrix $Y$. More details can be found in Appendix B.

For thresholding + LSAP, we ensure that the detected subgraphs are of equal size and have a one-to-one node-node mapping (to validate their isomorphism) by running the Hungarian algorithm on the remaining rows and columns of $Y$ after thresholding. We cannot run LSAP on the original $Y$ since LSAP would select all nodes in the smaller of the two graphs.

Neither of these simpler subgraph extraction methods enforces the subgraph isomorphism constraint, which explains their worse performance compared with GSE.

We find that our major novelties (GSE and normalization technique) are the most important components in producing good performance.

## G    SCALABILITY STUDY

is trained on graph pairs with ground-truth MCS results, which can be either obtained by MCSPLIT or the following way which provides cheap supervision without using any exact MCS solver. The high level idea of generating ground-truth MCS training pairs without exact MCS solver is to create such pairs with a smart design instead of computing MCS for any given pair of graphs. One possible way to create such pair is to extract an induced subgraph from a given graph, and the ground-truth MCS of the two graphs is naturally the extracted subgraph. More concretely, our experimental setup is as follows: We generate training graph pairs by first using the BarabsiAlbert model (Barabási & Albert, 1999) to generate 1000 graphs of size 32. For each generated graph, We randomly extract one connected 16-node subgraph from it. Each generated graph and extracted subgraph form one pair, giving a total of 1000 training graph pairs (our training set). Notice, this generation procedure allows us to know the MCSs during generation.

---

[4]IMAGE-PCA performs an additional normalization and adopts a slightly different Sinkhorn Layer for Linear Assignment which yields in similar performance to GMN + Sinkhorn Softmax as indicated in Table 1.

We follow a similar procedure for testing set, where we use the Barabsi-Albert model to generate 100 graphs of size 16, 32, 64, and 128 (denoted as "Test Dataset Size" in the table below). For each generated graph, we extract one connected 8-, 16-, 32-, and 64- node subgraph respectively. This gives us 5 test sets, each with 100 graph pairs.

We use the following 5 metrics for thorough evaluation:

1. **Solved %**: It measures the percentage of pairs that the model can successfully finish within 100 seconds.

2. **Soft %**: It measures the fraction of the predicted MCS size over the true MCS size for the isomorphic extracted subgraphs (Appendix D).

3. **Iso %**: Among the pairs that can be solved within the time budget, it measures the percentage of pairs whose extracted subgraphs are isomorphic. This is an important metric because subgraph isomorphism is a key constraint required by the definition of MCS.

4. **Dev in # nodes**: among the pairs that can be solved within the time budget, it measures the average deviation of the number of nodes in the predicted MCS versus the number of nodes in the true MCS. The range of this metric is $[0, N]$ where N is the number of nodes of the largest graph in a dataset. This metric gives a more intuitive understanding of the performance of a model compared to "Soft %" since it reports the number of nodes directly.

5. **(Average) Runtime (msec)**: It measures the average running time per testing pairs that the model solves within the time budget. In other words, if a model fails at solving a pair within the time budget, the runtime will NOT be taken into account by this metric for fair comparison

We set the time budget to 100 seconds and 500 seconds for MCSPLIT respectively, and the results are shown in Table 4.

Table 4: Scalability study results on AIDS.

| Test Dataset size | Metrics | MCSPLIT (100s) | MCSPLIT (500s) | MCSPLIT |
|---|---|---|---|---|
| 16 | Solved % | **100.000** | **100.000** | **100.000** |
| | Soft % | **100.000** | **100.000** | 99.625 |
| | Iso % | **100.000** | **100.000** | **100.000** |
| | Dev in # nodes | **0** | **0** | 0.030 |
| | Runtime (msec) | 295.576 | **236.502** | 550.688 |
| 32 | Solved % | **100.000** | **100.000** | **100.000** |
| | Soft % | **100.000** | **100.000** | 99.563 |
| | Iso % | **100.000** | **100.000** | **100.000** |
| | Dev in # nodes | **0** | **0** | 0.070 |
| | Runtime (msec) | **333.793** | 340.310 | 787.901 |
| 64 | Solved % | 61.000 | 62.000 | **100.000** |
| | Soft % | 61.000 | 62.000 | **98.843** |
| | Iso % | **100.000** | **100.000** | **100.000** |
| | Dev in # nodes | **0** | **0** | 0.370 |
| | Runtime (msec) | 4509.056 | 10351.813 | **940.581** |
| 128 | Solved % | 26.000 | 28.000 | **100.000** |
| | Soft % | 26.000 | 28.000 | **75.484** |
| | Iso % | **100.000** | **100.000** | **100.000** |
| | Dev in # nodes | **0** | **0** | 15.690 |
| | Runtime (msec) | 1220.117 | 18809.848 | **1194.089** |

NEURALMCS achieves performance close to the exact ground truth solver in terms of accuracy, and can scale to much larger graphs which the exact solver fails, most of the time, at solving within the time budget.

For accuracy, we see that the size of MCS extracted for the model is often over 98% the true MCS size. Interestingly, for graphs of size 64, we find that the extracted subgraphs differ between our model and the true MCS only in less than one node (0.37 nodes) on average. This indicates that our proposed method (NEURALMCS) can detect MCS that the current state of the art (MCSPLIT) cannot, due to our careful design and the incorporation of learning algorithms.

For running time, our model is comparable to or slower than the ground truth detector for simpler cases (¡ 1 second) but much faster for larger graphs. This could be due to our model incurring overhead from Python implementation, while MCSPLIT is implemented in C++. However, when the dataset size equals 128, MCSPLIT fails for most pairs, as the MCS problem is NP-hard, while NEURALMCS can solve all the pairs due to guaranteed time complexity (Section 3.4). Notice, there is no theoretical time complexity guarantee for MCSPLIT (exponential time complexity in the worst case (McCreesh et al., 2017)), resulting in significant average running time increase from 1220.1 msec to 18809.8 msec when only 2 additional pairs are solved by increasing the time budget from 100 seconds to 500 seconds. In fact, we observed that the actual running time of the branch-and-bound algorithm MCSPLIT strongly depends on the actual graph structures varying from graph to graph.

In summary, when graphs are larger and larger, MCSPLIT quickly becomes almost unusable in practice due to an inability to yield results most of the time, while NEURALMCS still extracts high-accuracy MCSs consistently and runs much faster than MCSPLIT with guaranteed theoretical time complexity. In practice, we observe that our model can run successfully on a 12-GB GPU until 4000-node graphs when the GPU runs out of memory, in which case MCSPLIT almost surely cannot be used either.

## H    DISCUSSION ON SUBGRAPH ISOMORPHISM CHECKING OF GUIDED SUBGRAPH EXTRACTION

In the proposed GUIDED SUBGRAPH EXTRACTION (GSE) strategy (Section 3.3.1), we check if the inclusion of a new node pair would result in two isomorphic subgraphs by checking if $||\boldsymbol{w}_1 - \boldsymbol{w}_2||$ is smaller than or equal to a threshold, which is a criteria that allows us to punish mismatched nodes more softly. While an iterative check would achieve efficiency benefits by avoiding computing the subgraph embeddings, it assumes at every step we have performed a non-ambiguous matching. For example, suppose we have 2 graphs, where in the current iteration of GSE, we have 2 fully connected 3-node subgraphs currently extracted with node-node mappings. We label the node ids of these 2 graphs as 1-2-3 and a-b-c respectively and the mappings as 1-a, 2-b, 3-c. If we select a new node 4 from $\mathcal{G}_1$ and a new node d from $\mathcal{G}_2$ and 4 is connected to 2 and d is connected to a, an iterative procedure would not be able to tell that the addition of node 4 and d to the MCS is valid (since 2-b is matched but NOT 2-a). Fundamentally, the structure for 1,2,3 and a,b,c are similar, so it is uncertain (and expected) that the computed node matchings would be 2-a, 1-b or 2-b, 1-a.

In contrast, our proposed checking strategy can account for the above-mentioned uncertainty by doing aggregation of node embeddings to obtain subgraph-level embeddings $w_1$ and $w_2$, since this does not involve explicitly finding the node-node mappings. Instead, it checks the isomorphism in a more principled way borrowing insights from Weisfeiler-Lehman (WL) graph isomorphism test (Shervashidze et al., 2011). In WL, each node in the two graphs is represented as an aggregation of local node features, and each graph is represented as an aggregation of node labels. The algorithm stops and decides two graphs are not isomorphic when the two graph-level node label sets are different. In our model, the graph-level node label set is equivalent to $\boldsymbol{w}_1$ and $\boldsymbol{w}_2$, and our GSE decides that the subgraphs are not isomorphic when the two graph-level representations are different enough.

In conclusion, our proposed checking strategy addresses the ambiguous node mapping issue which the iterative isomorphism check could not solve, and is theoretically connected to the WL algorithm for graph isomorphism test.

## I    DISCUSSION ON LEARNINING CAPACITY OF GUIDED SUBGRAPH EXTRACTION

In our current model, we form a matching matrix for both training and inference, and perform our loss function on the matching matrix during training (Section 3.2), and utilize GSE (Section 3.3) guided by the matching matrix during inference to extract subgraphs. We use GMN for node embeddings (Section 2.2 and Section 3.1) and feed the matching matrix into GSE, which is fixed during the iterative GSE process. However, one potential issue is that once a node pair is selected, the extracted subgraph grows by one node causing the candidate pairs to change in the next iteration.

Ideally, node embeddings should be updated to reflect such change after each iteration. By introducing learnable components to iteratively update the node embeddings to reflect such change, we can train the GSE component using guidance from the ground-truth node mappings in the MCS, making the training stage and inference stage consistent and match each other.

To accomplish this, we propagate the node embeddings at each iteration of GSE with learnable weights to recompute the matching matrix $Y$ such that the next iteration's node embeddings will be conditionally updated based on the current extracted subgraph. To achieve it, we extend GMN to update the node embeddings for the extracted subgraph at each GSE iteration. GMN updates the node embeddings of two graphs jointly by performing intra- and inter-graph message passing. Thus, one can directly apply GMN to the extracted subgraph at each GSE iteration, and calculate the loss function at the end of GSE process (replacing the current BCE loss on the matching matrix $Y$), achieving conditional node embeddings in the GSE step of the model.

In implementation, we make a further modification to GMN by not propagating to matched nodes in the two extracted subgraphs. This is because we want the node embeddings for the currently extracted isomorphic subgraphs to stay as consistent as possible and not be influenced by any unpicked nodes in the larger graph or any nodes from the opposing graph.

We run both modified versions (with and without further modification) of NEURALMCS on AIDS, and the performance increase is only marginal ($< 1\%$ increase). Therefore, by forgoing this step during training and only using GSE only during testing, we can gain a free speed up in training time.

## J    MORE CASE STUDY

All case study plots can be seen in Figure 5. We see that our model is able to differentiate difficult input pairs, where adding any extra nodes would break the MCS constraints. In the LINUX, IMDB, and REDDIT dataset, we see examples where the graph structures are vastly different, yet NEURALMCS is still able to correctly differentiate MCS nodes. In the AIDS dataset, we see the model is able to successfully extract subgraphs which maintain node labels. In the IMDB dataset, we see the model can handle denser and larger size graphs.

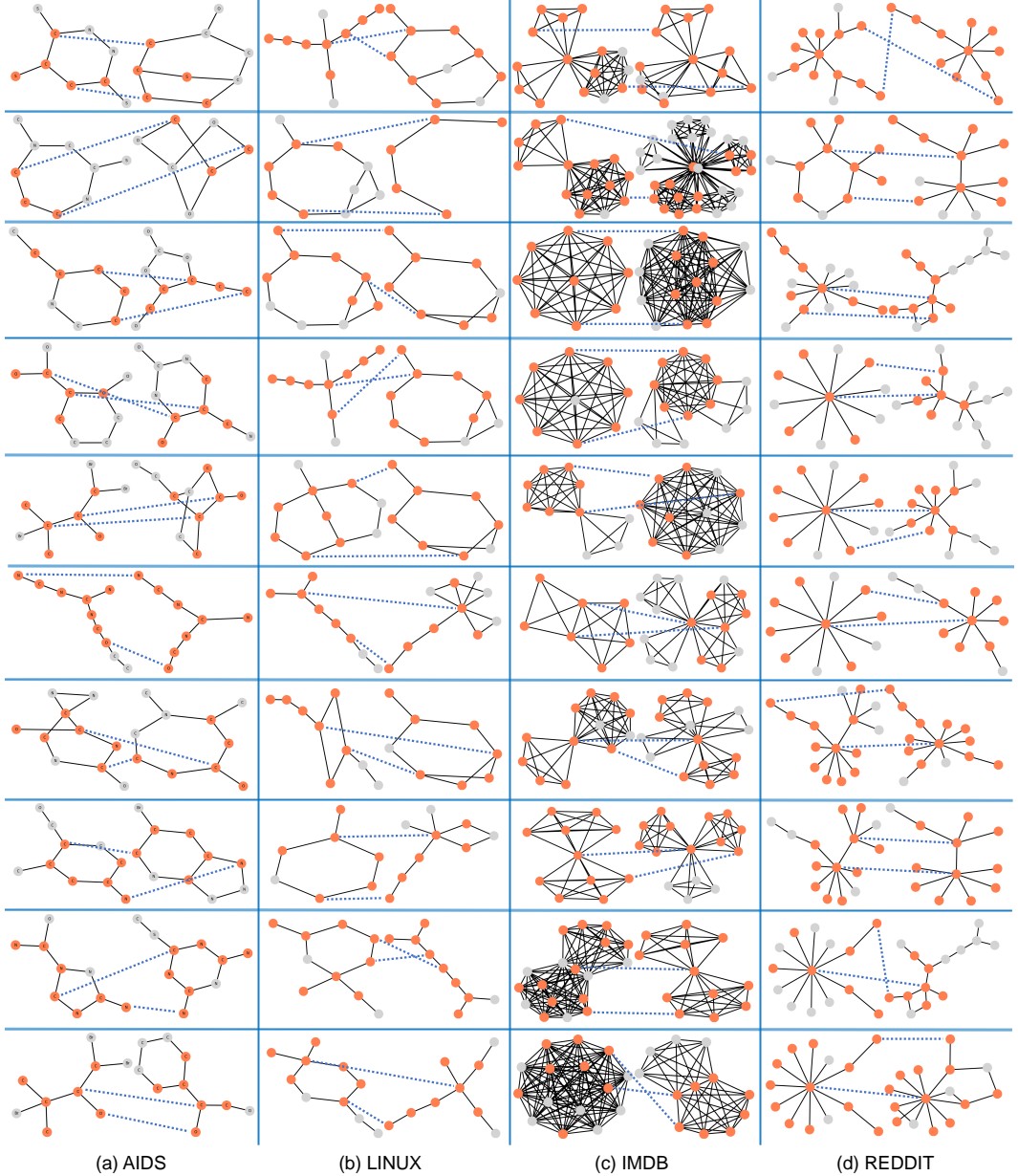

Figure 5: Case study.

