# OpenReview forum: "Neural Maximum Common Subgraph Detection with Guided Subgraph Extraction"
_ICLR.cc/2020/Conference — Reject_

### Official Review · AnonReviewer3 · 2019-10-23
**Official Blind Review #3**

**Rating:** 3

**Review:**

This paper proposes a novel algorithm NeuralMCS for maximum common subgraph (MCS) identification. The proposed algorithm consists of two components. One is a neural-network model based on Graph Matching Networks (GMN, Li et al, 2019) to learn a node-to-node matching matrix from examples of the ground-truth MCS results. Another is the algorithm called GSE (Guided Subgraph Extraction) to obtain an MCS by making an explicit assignment from the estimated matching matrix by the NN model. Experimental comparisons are made to other NN-based approaches combined with threshold-based assignments by the Hungarian algorithm (w.r.t the accuracy) and to a state-of-the-art exact algorithm MCSplit, and show the effectiveness of NeuralMCS.

This paper proposes an interesting method for MCS detection which would have large application interests. Though the basic idea is nice,  the reported performance gains would be a bit less convincing due to the following evaluation problem and its weak novelty.

The algorithm has two parts, and the first NN part to learn a matching matrix is mostly based on the already existing algorithm of GMN (Li et al, 2019). The novel part would be primarily in post-processing normalization (described in 3.1) for the matching matrix and seems to also be applicable to other NNs (for example, GAT?). The second part GSE to get an explicit subgraph also seem to be applied independently to the first part. We can see that combining these two parts worked, but it is unclear how each component contributes to the performance gain compared to any possible alternatives of each part.

I understand that MCS detection from a matching matrix (and node state vectors) is not exact if we just use Hungarian-like linear assignment problem (LAP) solvers for a submatrix obtained by a simple thresholding, but both post-processing normalization and GSE parts (which brings the novelty) can be more carefully evaluated through some 'ablation studies' using some simple alternative substitutes.

**Experience Assessment:**

I have published one or two papers in this area.

**Review Assessment: Checking Correctness Of Derivations And Theory:**

N/A

**Review Assessment: Checking Correctness Of Experiments:**

I carefully checked the experiments.

**Review Assessment: Thoroughness In Paper Reading:**

I read the paper thoroughly.

---

> ### Author Response · Authors · 2019-11-10
> **Response to Review #3**
>
> Thank you for your helpful comments!
>
> As pointed out, we first form a matching matrix then extract a subgraph guided by this matrix. To form the matching matrix, we utilize representation learning to make node embeddings; compute X using similarity scores from node embeddings (Section 3.1, paragraphs 3 and 4); compute Y through normalization of X (Section 3.1; Equations 1 and 2). To perform extraction, we utilize the GSE method proposed (Section 3.3).
>
> Regarding the evaluation concern, we performed more in-depth ablation studies as shown below to show the importance of each component on the AIDs dataset (the numbers are the Exact % defined in Section 4.1):
> +----------------------------------------------------------+------------------------------------------------------+
> |         Matching Matrix                                     |         Subgraph Extraction Strategy         |
> |           Computation                                         +------------+----------------+-----------------------+
> |                                                                            |     GSE    | Threshold | Threshold + LAP |
> +----------------------------------------------------------+------------+----------------+-----------------------+
> | GMN + Our Normalization (NeuralMCS)    |   90.692  |    26.057    |          17.339         |
> | GAT + Our Normalization                              |   87.119  |    21.206    |          16.159         |
> | GMN + Sigmoid (BasicGMN)                         |   31.596  |    10.521    |          12.488        |
> | GMN + Sinkhorn Softmax                              |   21.861  |    12.488    |           0.197         |
> +----------------------------------------------------------+-------------+---------------+-----------------------+
>
> GMN + Our Normalization is the proposed NeuralMCS architecture.
>
> 1) On the importance of GMN:
>
> GAT + Our Normalization uses GAT to encode the initial node embeddings and feed these embeddings into the NeuralMCS pipeline instead of the GMN node embeddings. As GAT does not perform inter-graph attention, the embeddings for one graph are not informed of the structure of the graph it is being matched to. This leads to lower performance.
>
> 2) On the importance of Normalization:
>
> GMN + Sigmoid (BasicGMN) uses sigmoid normalization, $y=1/(1+e^{-x})$, on each individual element of the X matrix, instead of our normalization scheme (Section 3.1; Equations 1 and 2) to obtain Y. As sigmoid treats each node-node pair in Y as independent (an incorrect assumption), we see that our normalization procedure performs the best.
>
> GMN + Sinkhorn Softmax uses successive row- and column-wise softmax normalization (softmax to ensure that the matching matrix Y is in the range (0,1)) on the X matrix (similar to Image-PCA’s [1] sinkhorn algorithm [2]) instead of our normalization scheme (Section 3.1; Equations 1 and 2). As softmax does not explicitly allow nodes to go unmatched (Section 3.1 paragraph 4), as dictated by the MCS definition, we see that our normalization procedure performs the best.
>
> 3) On the importance of GSE:
>
> For each Matching Matrix Computation method, we run 3 different subgraph extraction strategies: GSE (Section 3.3), thresholding, and thresholding + LAP (i.e. Hungarian Algorithm, described in Appendix B paragraph 3).
>
> For thresholding, for each of the two graphs, we select the nodes whose probabilities of being included in the MCS are greater than a tunable threshold, yielding two subgraphs. We calculate such probabilities by taking the summation of rows and columns of the matching matrix Y. More details can be found in Appendix B, the third paragraph.
>
> For thresholding + LAP, we ensure that the detected subgraphs are of equal size and have a one-to-one node-node mapping (to validate their isomorphism) by running the Hungarian algorithm [3] on the remaining rows and columns of Y after thresholding. We cannot run LAP on the original Y since LAP would select all nodes in the smaller of the two graphs.
>
> Neither of these simpler subgraph extraction methods enforces the subgraph isomorphism constraint, which explains their worse performance compared with GSE detailed in Section 3.3.
>
> We find that our major novelties (GSE and normalization technique) are the most important components in producing good performance.
>
> We appreciate your valuable feedback again!
>
> [1] Wang, Runzhong, Junchi Yan, and Xiaokang Yang. "Learning Combinatorial Embedding Networks for Deep Graph Matching." ICCV (2019).
> [2] Knight, Philip A. "The Sinkhorn–Knopp algorithm: convergence and applications." SIAM Journal on Matrix Analysis and Applications 30.1 (2008): 261-275.
> [3] Kuhn, Harold W. "The Hungarian method for the assignment problem." Naval research logistics quarterly 2.1‐2 (1955): 83-97.

---

### Official Review · AnonReviewer1 · 2019-11-05
**Official Blind Review #1**

**Rating:** 6

**Review:**

The authors proposed a novel method to find the Maximum Common Subgraph (MCS) of two graphs. I am familiar with the quadratic assignment problem (QAP) based graph matching and I am not very familiar with the MCS problem.

The authors adopt Graph Matching Networks (GMN) for feature embedding, and then similarity matrix X can be generated by computing the similarities between the embeddings. The similarity matrix is then normalized using a similar way as the sinkhorn procedure in [1-3]. The Assignment matrix then can be given from X. Then a novel procedure, named Guided subgraph Extraction (GSE, which is considered as the main contribution of this paper), is used  to get an MCS from assignment matrix. Here the authors may consider a simple baseline, which is to use QAP to give the assignment matrix, and then run GSE to obtain the MCS.

Overall the paper is well written, and the experiment is good and solid.

Some suggestions:
The GCN based GMN might not be the best choice for graph embedding. The authors may consider stronger Graph Neural Networks such as DGCNN (used in [3]) or Message Passing Neural Network (used in [4] and [5]) as the graph embedding module in the future work.

[1] Deep Learning of Graph Matching, CVPR18
[2] Learning Combinatorial Embedding Networks for Deep Graph Matching. ICCV19
[3] Deep Closest Point: Learning Representations for Point Cloud Registration. ICCV19
[4] Deep Graphical Feature Learning for the Feature Matching Problem, ICCV19
[5] Neural Message Passing for Quantum Chemistry, ICML17

**Experience Assessment:**

I have published in this field for several years.

**Review Assessment: Checking Correctness Of Derivations And Theory:**

I assessed the sensibility of the derivations and theory.

**Review Assessment: Checking Correctness Of Experiments:**

I assessed the sensibility of the experiments.

**Review Assessment: Thoroughness In Paper Reading:**

I read the paper at least twice and used my best judgement in assessing the paper.

---

> ### Author Response · Authors · 2019-11-09
> **Response to Review #1**
>
> Thank you for your constructive feedback!
>
> As you mentioned, the construction of matching matrix, Y (Section 3.1 paragraph 2), can be approached in 2 ways:
> - directly compute Y using some QAP-based approach
> - utilize representation learning to form node embeddings; compute similarity matrix, X, using similarity scores from node embeddings (Section 3.1 paragraphs 3 and 4); compute Y through normalization of X (Section 3.1; Equations 1 and 2).
>
> We consider alternative methods and suggestions of constructing the matching matrix, Y, whose results on AIDS are shown below:
> +-----------------------------------------------------------------+--------------+-------------+
> |         Matching Matrix Computation                      |  Exact %  |   Soft %   |
> +-----------------------------------------------------------------+--------------+-------------+
> |     GMN + Our Normalization (NeuralMCS)         |   90.692   |   97.899   |
> |     DGCNN + Our Normalization                            |   86.169   |   96.770   |
> |     GW-QAP                                                                |   23.894   |   56.150   |
> +-----------------------------------------------------------------+-------------+--------------+
>
> GMN + Our Normalization: It is the proposed NeuralMCS architecture.
>
> DGCNN + Our Normalization: It utilizes DGCNN [1] (adopted in [2]) to form node embeddings; computation of X and Y from these node embeddings are the same as in NeuralMCS.We find that the representation learning approach significantly improves the quality of the Y matrix. While DGCNN achieves reasonable performance, we see that GMN outperforms it as GMN uses inter-graph message passing capturing the graph-graph matching relation.
>
> GW-QAP: It utilizes QAP [3] to directly obtain the Y matrix. This method produces an assignment matrix which is treated as Y by solving for the Gromov-Wasserstein (GW) discrepancy [4] for 2 graphs. The assignment matrix is soft, which is [3] is further turned into hard node-node mappings while we directly feed the soft matrix into the GSE step. The method is denoted as GW-QAP to be specific. Notice it directly solves the optimization problem without learnable components adaptive to the MCS ground-truth node-node mappings, so the result is relatively poor.
>
> In summary, we find that GMN outperforms DGCNN and QAP-based approaches for producing the matching matrix Y.
>
> We would like to thank you again for your helpful suggestions.
>
> [1] Wang, Yue, et al. "Dynamic graph cnn for learning on point clouds." ACM Transactions on Graphics (TOG) 38.5 (2019): 146.
> [2] Wang, Yue, and Justin M. Solomon. "Deep Closest Point: Learning Representations for Point Cloud Registration." ICCV (2019).
> [3] Xu, Hongteng, Dixin Luo, and Lawrence Carin. "Scalable Gromov-Wasserstein Learning for Graph Partitioning and Matching." NeurIPS (2019).
> [4] Mémoli, Facundo. "Gromov–Wasserstein distances and the metric approach to object matching." Foundations of computational mathematics 11.4 (2011): 417-487.

---

### Official Review · AnonReviewer2 · 2019-11-08
**Official Blind Review #2**

**Rating:** 3

**Review:**

This paper proposed a graph net based approach for subgraph matching. The general idea is based on the graph matching network (Li et.al, ICML 2019) that computes the node embeddings of two graphs with co-attentions. The training requires the supervision of ground truth matching. During inference an iterative method with heuristic stopping criteria is used. Experiments on tiny graphs show better results than learning based baselines, but worse results than MCS solver.

Overall the paper is well motivated. However there are several major concerns with the paper:

1. Since it relies on the solver to provide training data, it might be hard to train on large graphs as there would be no cheap supervision. Also it seems that getting slightly faster but much worse results than the solver on small graphs is not that exciting.

2. It seems there's a mismatch between training and inference. The inference method is done iteratively, where the Eq (6) is somewhat not clear to me: as this ||w1-w2|| criteria is not trained during training, it seems quite heuristic by doing so.

3. I’m not sure why the two stop conditions are needed. One can easily check (incrementally) whether the added nodes are isomorphic.

4. The graphs used in experiments are too small.

Some other minor issues:

It would be better to define Y with Eq (1) and Eq (2) in the paper. There seems to be no explicit definition of Y.



**Experience Assessment:**

I have published in this field for several years.

**Review Assessment: Checking Correctness Of Derivations And Theory:**

N/A

**Review Assessment: Checking Correctness Of Experiments:**

I assessed the sensibility of the experiments.

**Review Assessment: Thoroughness In Paper Reading:**

I read the paper thoroughly.

---

> ### Author Response · Authors · 2019-11-12
> **Response to Review #2 (Part 1)**
>
> First of all, we would like to sincerely thank you for your feedback and suggestions.
>
> Regarding the concerns:
>
> 1a) Regarding cheap supervision, we can indeed generate ground-truth MCS training pairs without using any exact MCS solver, as demonstrated in the new experiment described below. Also, we would like to point out that our model can be trained on smaller graph pairs with cheap supervision and tested on other (larger) graph pairs.
>
> The high level idea of generating ground-truth MCS training pairs without exact MCS solver is to create such pairs with a smart design instead of computing MCS for any given pair of graphs. One possible way to create such pair is to extract an induced subgraph from a given graph, and the ground-truth MCS of the two graphs is naturally the extracted subgraph.  More concretely, our experimental setup is as follows:
>
> We generate training graph pairs by first using the Barabási–Albert model [1] to generate 1000 graphs of size 32. For each generated graph, We randomly extract one connected 16-node subgraph from it. Each generated graph and extracted subgraph form one pair, giving a total of 1000 training graph pairs (our training set). Notice, this generation procedure allows us to know the MCSs during generation.
>
> We follow a similar procedure for testing set, where we use the Barabasi-Albert model to generate 100 graphs of size 16, 32, 64, and 128 (denoted as “Test Dataset Size” in the table below). For each generated graph, we extract one connected 8-, 16-, 32-, and 64- node subgraph respectively. This gives us 5 test sets, each with 100 graph pairs.
>
> We use the following 5 metrics for thorough evaluation:
> Solved %: It measures the percentage of pairs that the model can successfully finish within 100 seconds.
> Soft %: It measures the fraction of the predicted MCS size over the true MCS size for the isomorphic extracted subgraphs (Section D paragraph 5).
> Iso %: Among the pairs that can be solved within the time budget, it measures the percentage of pairs whose extracted subgraphs are isomorphic. This is an important metric because subgraph isomorphism is a key constraint required by the definition of MCS.
> Dev in # nodes: among the pairs that can be solved within the time budget, it measures the average deviation of the number of nodes in the predicted MCS versus the number of nodes in the true MCS. The range of this metric is [0,N] where N is the number of nodes of the largest graph in a dataset. This metric gives a more intuitive understanding of the performance of a model compared to “Soft %” since it reports the number of nodes directly.
> (Average) Runtime (msec): It measures the average running time per testing pairs that the model solves within the time budget. In other words, if a model fails at solving a pair within the time budget, the runtime will NOT be taken into account by this metric for fair comparison
>
> We set the time budget to 100 seconds and 500 seconds for McSplit (state-of-the-art exact solver) [2] respectively, and the results are as follows:

---

> > ### Author Response · Authors · 2019-11-12
> > **Response to Review #2 (Part 2)**
> >
> >
> > +------------------------+-----------------------+--------------------+---------------------+-----------------+
> > | Test Dataset size |        Metrics        | McSplit (100s) | McSplit (500s) | NeuralMCS |
> > +------------------------+-----------------------+--------------------+---------------------+-----------------+
> > |             16              | Solved %             |            100.000 |            100.000 |        100.000 |
> > |                               | Soft %                  |            100.000 |            100.000 |          99.625 |
> > |                               | Iso %                    |           100.000 |             100.000 |        100.000 |
> > |                               | Dev in # nodes   |                       0 |                        0 |            0.030 |
> > |                               | Runtime (msec) |            295.576 |            236.502 |        550.688 |
> > +------------------------+-----------------------+--------------------+---------------------+-----------------+
> > |             32              | Solved %             |            100.000 |            100.000 |        100.000 |
> > |                               | Soft %                  |            100.000 |            100.000 |          99.563 |
> > |                               | Iso %                    |            100.000 |            100.000 |        100.000 |
> > |                               | Dev in # nodes   |                       0 |                        0 |            0.070 |
> > |                               | Runtime (msec) |            333.793 |            340.310 |        787.901 |
> > +------------------------+-----------------------+--------------------+---------------------+-----------------+
> > |             64              | Solved %             |              61.000 |              62.000 |        100.000 |
> > |                               | Soft %                  |              61.000 |              62.000 |          98.843 |
> > |                               | Iso %                    |           100.000 |             100.000 |        100.000 |
> > |                               | Dev in # nodes   |                      0 |                         0 |            0.370 |
> > |                               | Runtime (msec) |         4509.056 |         10351.813 |       940.581 |
> > +------------------------+-----------------------+--------------------+---------------------+-----------------+
> > |            128             | Solved %             |             26.000 |               28.000 |        100.000 |
> > |                               | Soft %                  |             26.000 |                28.000 |         75.484 |
> > |                               | Iso %                    |           100.000 |             100.000 |        100.000 |
> > |                               | Dev in # nodes   |                      0 |                         0 |          15.690 |
> > |                               | Runtime (msec) |         1220.117 |         18809.848 |     1194.089 |
> > +------------------------+-----------------------+--------------------+---------------------+-----------------+

---

> > > ### Author Response · Authors · 2019-11-12
> > > **Response to Review #2 (Part 3)**
> > >
> > > As shown in the table, NeuralMCS achieves performance close to the exact ground truth solver in terms of accuracy, and can scale to much larger graphs which the exact solver fails, most of the time, at solving within the time budget.
> > >
> > > For accuracy, we see that the size of MCS extracted for the model is often over 98% the true MCS size. Interestingly, for graphs of size 64, we find that the extracted subgraphs differ between our model and the true MCS only in less than one node (0.37 nodes) on average. This indicates that our proposed method (NeuralMCS) can detect MCS that the current state of the art (McSplit) cannot, due to our careful design and the incorporation of learning algorithms.
> > >
> > > For running time, our model is comparable to or slower than the ground truth detector for simpler cases (< 1 second) but much faster for larger graphs. This could be due to our model incurring overhead from Python implementation, while MCSplit is implemented in C++. However, when the dataset size equals 128, McSplit fails for most pairs, as the MCS problem is NP-hard, while NeuralMCS can solve all the pairs due to guaranteed time complexity (Section 3.4). Notice, there is no theoretical time complexity guarantee for McSplit (exponential time complexity in the worst case [2]), resulting in significant average running time increase from 1220.1 msec to 18809.8 msec when only 2 additional pairs are solved by increasing the time budget from 100 seconds to 500 seconds. In fact, we observed that the actual running time of the branch-and-bound algorithm McSplit strongly depends on the actual graph structures varying from graph to graph.
> > >
> > > In summary, when graphs are larger and larger, McSplit quickly becomes almost unusable in practice due to an inability to yield results most of the time, while NeuralMCS still extracts high-accuracy MCSs consistently and runs much faster than McSplit with guaranteed theoretical time complexity. In practice, we observe that our model can run successfully on a 12-GB GPU until 4000-node graphs when the GPU runs out of memory, in which case McSplit almost surely cannot be used either.
> > >
> > > [1] Barabási, Albert-László, and Réka Albert. "Emergence of scaling in random networks." science 286.5439 (1999): 509-512.
> > > [2] McCreesh, Ciaran, Patrick Prosser, and James Trimble. "A partitioning algorithm for maximum common subgraph problems." IJCAI (2017): 712-719.

---

> > > > ### Author Response · Authors · 2019-11-12
> > > > **Response to Review #2 (Part 4)**
> > > >
> > > > 1b) Regarding the performance of our model, we have implemented an easy-to-implement strategy to further boost our performance. During GSE (Algorithm 1 and Section 3.3), instead of selecting just the node with the highest matching score (argmax of the masked matching matrix; Equation 7), we check all the unmatched neighbors to the currently extracted subgraph and select the node which both has the highest matching score and preserves subgraph isomorphism (Algorithm 1 line 10). This strategy indeed incurs additional running time, but, in practice, is still around 10x faster than the exact solver. With this small modification (trick), NeuralMCS+trick is able to reach near-perfect performance on all datasets, as shown below (on AIDS, LINUX, IMDB, and REDDIT):
> > > >
> > > > On AIDS:
> > > > +---------------------------------+-------------+-----------+----------------------------+
> > > > |              Method               | Exact %  |  Soft %  | Running Time [ms] |
> > > > +---------------------------------+-------------+-----------+----------------------------+
> > > > | NeuralMCS                      |    90.692 |  97.899  |                 73.057       |
> > > > | NeuralMCS + trick          |    98.525 |  99.626  |                  80.054      |
> > > > | McSplit                             |  100.000 | 100.000 |             1404.844       |
> > > > +---------------------------------+-------------+-----------+----------------------------+
> > > >
> > > > On LINUX:
> > > > +---------------------------------+-------------+-----------+----------------------------+
> > > > |              Method               | Exact %  |  Soft %  | Running Time [ms] |
> > > > +---------------------------------+-------------+-----------+----------------------------+
> > > > | NeuralMCS                      |    93.779 |  98.926  |                131.053      |
> > > > | NeuralMCS + trick          |    99.674 |  99.955  |                117.140      |
> > > > | McSplit                             |  100.000 | 100.000 |             1288.319       |
> > > > +---------------------------------+-------------+-----------+----------------------------+
> > > >
> > > > On IMDB:
> > > > +---------------------------------+-------------+-----------+----------------------------+
> > > > |              Method               | Exact %  |  Soft %  | Running Time [ms] |
> > > > +---------------------------------+-------------+-----------+----------------------------+
> > > > | NeuralMCS                      |    83.734 |  92.366  |                 59.570       |
> > > > | NeuralMCS + trick          |    97.235 |  99.613  |                226.657      |
> > > > | McSplit                             |  100.000 | 100.000 |              1862.181      |
> > > > +---------------------------------+-------------+-----------+----------------------------+
> > > >
> > > > On REDDIT:
> > > > +---------------------------------+-------------+-----------+----------------------------+
> > > > |              Method               | Exact %  |  Soft %  | Running Time [ms] |
> > > > +---------------------------------+-------------+-----------+----------------------------+
> > > > | NeuralMCS                      |    77.124 |  93.040  |                 98.284       |
> > > > | NeuralMCS + trick          |    96.078 |  99.562  |                142.112      |
> > > > | McSplit                             |  100.000 | 100.000 |             12843.25       |
> > > > +---------------------------------+-------------+-----------+----------------------------+
> > > >
> > > > Excitingly, on IMDB, the performance increases from 83.7% to 97.2% (13.5% increase!), and on REDDIT, the performance increases from 77.1% to 96.1% (19.0% increase!). These performance benefits are expected as the strategy makes NeuralMCS less greedy. We can view this trick as a tradeoff between accuracy and runtime and can be turned in a tunable hyperparameter. Notice, with or without the strategy, NeuralMCS still performs much faster than McSplit.
> > > >
> > > > In conclusion, with a trick we can obtain faster performance and near-perfect accuracy (>96% testing pairs are solved exactly correctly across the four datasets) compared to the solver, McSplit.
> > > >
> > > > Note: As mentioned in Section 3.4, the original NeuralMCS model runs in $O(s * |V_1| * |V_2|)$ time, where $s$ is the size of the predicted MCS, which in the worst case is $\min (|V_1|,|V_2|)$. The new trick results in $O(s * |V_1| * |V_2| * log(|V_1| * |V_2|))$ time complexity due to sorting the matching scores. Denote $V = \max (|V_1|,|V_2|)$. Then the original time complexity can be simplified to $O(V^3)$ and the new time complexity being $O(V^3 * log(V))$.
> > > >
> > > > ... response to concerns continued in part 5)

---

> > > > > ### Author Response · Authors · 2019-11-13
> > > > > **Response to Review #2 (Part 5)**
> > > > >
> > > > > 2a) Regarding the mismatch between training and testing, we tried over several modifications where loss are also put in the GSE step, but the performance enhancement is only marginal. The detailed design of such modifications is introduced below.
> > > > >
> > > > > In our current model, we form a matching matrix for both training and inference, and perform our loss function on the matching matrix during training (Section 3.2), and utilize GSE (Section 3.3) guided by the matching matrix during inference to extract subgraphs. We use GMN [3] for node embeddings (Section 2.2 and Section 3.1) and feed the matching matrix into GSE, which is fixed during the iterative GSE process. However, one potential issue is that once a node pair is selected, the extracted subgraph grows by one node causing the candidate pairs to change in the next iteration. Ideally, node embeddings should be updated to reflect such change after each iteration. By introducing learnable components to iteratively update the node embeddings to reflect such change, we can train the GSE component using guidance from the ground-truth node mappings in the MCS, making the training stage and inference stage consistent and match each other.
> > > > >
> > > > > To accomplish this, we smartly propagate the node embeddings at each iteration of GSE with learnable weights to recompute the matching matrix Y (so that the matching matrix Y evolves) such that the next iteration’s node embeddings will be conditionally updated based on the current extracted subgraph. The basic idea behind it is to extend GMN to update the node embeddings for the extracted subgraph at each GSE iteration. GMN updates the node embeddings of two graphs jointly by performing intra- and inter-graph message passing. Thus, one can directly apply GMN to the extracted subgraph at each GSE iteration, and calculate the loss function at the end of GSE process (replacing the current BCE loss on the matching matrix Y), achieving conditional node embeddings in the GSE step of the model.
> > > > >
> > > > > In implementation, we make a further modification to GMN by not propagating to matched nodes in the two extracted subgraphs. This is because we want the node embeddings for the currently extracted isomorphic subgraphs to stay as consistent as possible and not be influenced by any unpicked nodes in the larger graph or any nodes from the opposing graph.
> > > > >
> > > > > We run both modified versions (with and without further modification) of NeuralMCS on AIDS, and the performance increase is only marginal (<1% increase). Therefore, by forgoing this step during training and only using GSE only during testing, we can gain a free speed up in training time.

---

> > > > > > ### Author Response · Authors · 2019-11-13
> > > > > > **Response to Review #2 (Part 6)**
> > > > > >
> > > > > > 2b) and 3) Regarding the Equation 6 (stopping condition 2; Algorithm 1 line 10-11), the $||w_1-w_2||$ search criteria is necessary in that it allows us to punish mismatched nodes more softly. While an iterative check would achieve performance benefits, it assumes at every step we have performed a non-ambiguous matching. For example, suppose we have 2 graphs, where in the current iteration of GSE (Algorithm 1 line 6), we have 2 fully connected 3-node subgraphs currently extracted with node-node mappings. We label the node id’s of these 2 graphs as 1-2-3 and a-b-c respectively and the mappings as 1-a, 2-b, 3-c. If we select a new node 4 from G1 and a new node d from G2 and 4 is connected to 2 and d is connected to a, an iterative procedure would not be able to tell that the addition of node 4 and d to the MCS is valid (since 2-b is matched but NOT 2-a). Fundamentally, the structure for 1,2,3 and a,b,c are similar, so it is uncertain (and expected) that the computed node matchings would be 2-a, 1-b or 2-b, 1-a.
> > > > > >
> > > > > > In contrast, our proposed stopping condition can account for the above-mentioned uncertainty by doing aggregation of node embeddings to obtain subgraph-level embeddings $w_1$ and $w_2$, since this does not involve explicitly finding the node-node mappings. Instead, it checks the isomorphism in a more principled way borrowing insights from Weisfeiler-Lehman (WL) graph isomorphism test [4] (Section 3.3.1 paragraph 3). In WL, each node in the two graphs is represented as an aggregation of local node features, and each graph is represented as an aggregation of node labels. The algorithm stops and decides two graphs are not isomorphic when the two graph-level node label sets are different. In our model, the graph-level node label set is equivalent to $w_1$ and $w_2$, and our GSE stops when the two graph-level representations are different.
> > > > > >
> > > > > > In conclusion, our proposed stopping condition addresses the ambiguous node mapping issue which the iterative isomorphism check could not solve, and is theoretically connected to the WL algorithm for graph isomorphism test.
> > > > > >
> > > > > > Regarding the first stopping condition (Algorithm 1 line 7-8), it ensures that if there are not any nodes adjacent to the currently selected MCS nodes, we do not continue selecting. The second stopping condition (Algorithm 1 line 10-11) ensures that if such nodes lead to non-isomorphic subgraphs, we do not continue selecting. Because node embeddings are not inherently aware of connectivity, the first stopping condition is essential in that we do not select non-connected nodes.
> > > > > >
> > > > > > 4) As shown in the results for (1a), our model is able to generalize to graphs of size >100.
> > > > > >
> > > > > > On the minor issue about the definition of Y: Thank you for pointing out this issue. We will update our paper with this suggestion.
> > > > > >
> > > > > > At the end, please allow us to express our sincere gratitude for all your helpful comments and suggestions again!
> > > > > >
> > > > > > [1] Barabási, Albert-László, and Réka Albert. "Emergence of scaling in random networks." science 286.5439 (1999): 509-512.
> > > > > > [2] McCreesh, Ciaran, Patrick Prosser, and James Trimble. "A partitioning algorithm for maximum common subgraph problems." IJCAI (2017): 712-719.
> > > > > > [3] Li, Yujia, et al. "Graph Matching Networks for Learning the Similarity of Graph Structured Objects." ICML (2019).
> > > > > > [4] Shervashidze, Nino, et al. "Weisfeiler-Lehman graph kernels." JMLR 12.Sep (2011): 2539-2561.

---

### Author Response · Authors · 2019-11-14
**General Response**

To all reviewers:

Thank you for all your responses and valuable feedback. We have answered all the questions raised in the three reviews and provide a summary here.

First, we ran further ablation studies to confirm that the proposed novelties (Matching Matrix Computation + Guided Subgraph Extraction (GSE)) contribute to the significant performance boosting. Please refer to our response to Review #1 and Review #3 for details.

Second, we ran our model on larger synthetic datasets with cheap supervision to show the scalability of the proposed method. We find that NeuralMCS is able to generalize to graphs of size >100 nodes, under which scenario the branch-and-bound based exact MCS solver fails most of the time. Please refer to our response to Review #2 Part 1-3 for details.

Third, we show that with a simple strategy, NeuralMCS could achieve near-perfect accuracy in computing the exact MCS results on all four datasets. Please refer to our response to Review #2 Part 4 for details.

Last not but least, in our response to Review #2 Part 5-6, we address additional concerns regarding the design and performance of NeuralMCS in Review #2.

We sincerely appreciate the concerns and suggestions pointed out by all reviewers!

---

### Decision · Program_Chairs · 2019-12-19

**Decision:**

Reject

**Comment:**

This paper proposed graph neural networks based approach for subgraph detection. The reviewers find that the overall the paper is interesting, however further improvements are needed to meet ICLR standard:
1. Experiments on larger graph. Slight speedup in small graphs are less exciting.
2. It seems there's a mismatch between training and inference.
3. The stopping criterion is quite heuristic.